# Enrichment of Aldolase C Correlates with Low Non-Mutated IDH1 Expression and Predicts a Favorable Prognosis in Glioblastomas

**DOI:** 10.3390/cancers11091238

**Published:** 2019-08-23

**Authors:** Yu-Chan Chang, Hsing-Fang Tsai, Shang-Pen Huang, Chi-Long Chen, Michael Hsiao, Wen-Chiuan Tsai

**Affiliations:** 1Genomics Research Center, Academia Sinica, Taipei 115, Taiwan; 2Department of Pathology, Tri-Service General Hospital, National Defense Medical Center, Taipei 114, Taiwan; 3Department of Neurology, Po-Jen General Hospital, Taipei 105, Taiwan; 4Department of Pathology, Taipei Medical University Hospital, Taipei Medical University, Taipei 110, Taiwan; 5Department of Pathology, College of Medicine, Taipei Medical University, Taipei 110, Taiwan; 6Department of Biochemistry, College of Medicine, Kaohsiung Medical University, Kaohsiung 807, Taiwan

**Keywords:** aldolase C, IDH1, glioblastoma, prognostic

## Abstract

The aldolases family is one of the main enzymes involved in the process of glycolysis. Aldolase C (ALDOC), which belongs to the aldolase family, is found in normal brain tissue and is responsible for the repair of injured tissue. However, the role of ALDOC in glioblastoma remains unclear. In this study, we data-mined in silico databases to evaluate aldolase family members’ mRNA expression in glioblastoma patient cohorts for determining its prognostic values. After that, we also performed immunohistochemical stain (IHC) analysis to evaluate protein expression levels of ALDOC in glioblastoma tissues. From The Cancer Genome Atlas (TCGA) database analyses, higher mRNA expression levels in normal brain tissue compared to glioblastoma was observed. In addition, compared to low-grade glioma, ALDOC expression was significantly downregulated in high-grade glioblastoma. Besides, the expression level of ALDOC was associated with molecular subtypes of glioblastomas and recurrent status in several data sets. In contrast, aldolase A (ALDOA) and aldolase B (ALDOB) revealed no significant prognostic impacts in the glioblastoma cohorts. Furthermore, we also proved that ALDOC mRNA and protein expression inversely correlated with non-mutated IDH1 expressions in glioblastoma patient cohorts. Additionally, the concordance of low ALDOC and high non-mutated IDH1 expressions predicted a stronger poor prognosis in glioblastoma patients compared to each of above tests presented alone. The plausible ALDOC and IDH1 regulatory mechanism was further elucidated. Our results support high ALDOC expression in glioblastomas that might imply the mutated status of IDH1, less possibility of mesenchymal subtype, and predict a favorable prognosis.

## 1. Introduction

Glioblastoma is one of the WHO grade IV malignant glial tumors of the central nervous system (CNS) with an aggressive behavior [1]. The median survival time of glioblastoma patients is less than two years, despite standard therapies, including tumor resection, concomitant radiotherapy, or adjuvant chemotherapy plus temozolomide [2]. Recent studies have proven that some molecular analysis of isocitrate dehydrogenase 1/2 (IDH 1/2), TP53, alpha thalassemia/mental retardation syndrome X-linked (ATRX), and o-6-methylguanine-DNA methyltransferase (MGMT) are associated with therapeutic efficacy and prognosis [3]. Survival of glioblastoma patients is still poor; despite great efforts to better understand the underlying molecular mechanisms and develop new therapeutic regimens, most glioblastomas still reveal unfavorable survival time. To date, it is believed that primary brain tumor (PBT) is a multifactorial neoplasm rather than a single risk factor disease. According to The Cancer Genome Atlas (TCGA) and transcriptional profiling studies, glioblastoma is divided into three subtypes: Proneural, classical, or mesenchymal [4]. A parallel comparison of these two studies revealed particularly strong agreement in the gene signatures associated with proneural and mesenchymal subtypes [5]. A number of transcription factors—including STAT3 (signal transducer and activator of transcription 3), C/EBP-β (CCAAT-enhancer-binding protein-β), and, more recently, the transcriptional coactivator TAZ (transcriptional coactivator with PDZ-binding motif)—have been identified as important regulators of the mesenchymal phenotype in glioblastomas [5]. 

Enzymes of the aldolase family are involved in metabolism and glycolysis. They include three isoforms: Aldolase A (ALDOA), aldolase B (ALDOB), and aldolase C (ALDOC) [6]. Most aldolase proteins are detected in various non-neoplastic human tissues [6,7]. Aldolase converts fructose 1-6, bisphosphate to glyceraldehyde 3-phosphate (G3P) and Dihydroxyacetone phosphate (DHAP), so they control fatty acid synthesis and glycolysis to further affect the Warburg effect or oxidative phosphorylation (OXPHOs). Otherwise, overexpression of ALDOA has been proven to be associated with poor overall outcome in colorectal and prostatic cancers, as well as lung non-small cell carcinoma [8,9,10]. ALDOA has a hypoxia response element (HRE) in the promoter region. Our investigation showed that HIF-1α protein could be stable and that it formed a positive feedback loop in tumorigenesis. In addition, IGF1-PI3K axis could regulate the binding affinity between ALDOA with γ-actin. On the contrary, the roles of ALDOB and ALDOC seem inconstant in different human cancers. Several studies have indicated that ALDOB and ALDOC could suppress tumor invasive and metastatic potential in hepatocellular and oral squamous carcinomas [7,11]. Otherwise, ALDOB and ALDOC are viewed as poor prognostic factors in rectal and esophageal cancers, respectively [12,13,14]. According to recent reports, ALDOC plays a functional role in brain development, brain trauma, and cardiomyocytes. Up to now, the function of ALDOC expression has been inhibited in glial cell neoplasm and is still undetermined. Moreover, IDH1 has been recognized as a key prognostic factor for glioblastoma patients. Based on its genetic events, it could be correlated with patients’ survival rate or tumor progression. Furthermore, the IDH1 protein is also involved in Tricarboxylic acid (TCA) cycle, catalyzing the oxidative decarboxylation of isocitrate to produce alpha-ketoglutarate and CO_2_. Combining all evidences, we propose that IDH1 may trigger metabolic reprogramming depending on the expression level of ALDOC in brain tumors.

In this study, we screened microarray chips and TCGA data and found that ALDOC was lower expressed in mesenchymal subtype of glioblastomas. Furthermore, ALDOC was also associated with significantly increased survival time in glioblastoma patients, implying a tumor-suppressor factor in glioblastomas. In addition, our results also show that the expression of ALDOC was associated with IDH1 genomic alteration event in glioblastomas. Finally, the results of immunohistochemical stain (IHC) of ALDOC also revealed an inverse correlation with non-neoplastic IDH1 expression and optimal prognosis. 

## 2. Results

### 2.1. ALDOC mRNA Expression Level Inversely Correlated with IDH1 and Prolonged Survival Time in Glioblastoma Patients

IDH1 expression has been regarded as an independent prognostic factor for glioma patients’ survival rate [15]. We observed that the aldolase family may play a key role for glioma tumorigenesis. We collected information of survival time, MGMT methylated status, CIMP status, and IDH1 mutant status from TCGA glioblastoma cohort (Figure 1A). Unlike ALDOA and ALDOB, we observed that ALDOC inversely correlated with histology stages through heat map analysis of TCGA glioblastoma cohort (*p* < 0.0001, *n* = 538, Figure 1A,B). We further analyzed the different subtypes of glioblastomas with characteristic pathway activity and response to therapy. According to the transcriptome array analyses, we found that various expression levels changed among proneural, classical, and mesenchymal subtypes. The analysis of TCGA data revealed that ALDOC enriched in proneural subgroup of glioblastomas. The trend is consistent with CD44 (as the mesenchymal marker) and inversely correlated with OLIG2 (as a non-mesenchymal marker) (Appendix A). To validate the subtypes’ definitions, we classified various subtypes depending on available parameters in glioblastoma. We further observed that ALDOC significantly lowered mRNA expression in the glioblastoma compared with the non-tumor part, the non G-CIMP compared with the G-CIMP group, and unmethylated MGMT compared with MGMT methylated patients (*p* < 0.0001, *p* = 0.0019, and *p* = 0.0356, respectively) (Figure 1C). We further validated high ALDOC expression in IDH1 R132 mutant form compared with wild-type (*p* = 0.0279, Figure 1D). Besides, ALDOC showed an inversed trend of mutation status of IDH1 in glioblastomas (Spearmen rho = ~0.169, *p* = 0.001, Figure 1E).

### 2.2. High ALDOC mRNA Expression Level Correlated with Low Non-Mutated IDH1 Expression and Longer Survival Time in Gliomas

Due to the differences of therapeutic approaches in low-grade and high-grade gliomas, we screened the specific genetic events and aldolase family in low-grade glioma (LGG) and glioblastoma TCGA cohort. Similar to ALDOC in glioblastomas, our results showed a significantly inverse correlation between ALDOC and IDH1 mutation status in TCGA glioma cohort (TCGA_GBMLGG, *n* = 1119, Figure 2A). We also analyzed the expression level of ALDOC using several clinical parameters. The results showed higher ALDOC mRNA level significantly in gain of chromosome 7, combined with loss of chromosome 10 group or chromosome 1p/19q co-deletion group (*p* < 0.0001, *p* < 0.0001, respectively) (Figure 2B). The observation indicates that ALDOC may be the tumor suppressor gene, and is activated when one or more tumor suppressor genes are blocked in chromosome 1p/19q regions. We also observed a consistent trend with the glioblastoma cohort, namely, a low level of ALDOC expression in the unmethylated MGMT and IDH1 wild-type group (*p* < 0.0001, *p* < 0.0001, respectively) (Figure 2B). Moreover, ALDOC has statistical values associated with histological type and tumor grade (*p* < 0.0001, *p* < 0.0001, respectively) (Figure 2B). Furthermore, the detection of ALDOC expression was performed in solid tissue of normal, primary tumor, and recurrence tumor groups. Compared with solid normal tissue, significantly lower ALDOC expression was detected in recurrence tumor (*p* = 0.0001, Figure 2C). Through TCGA data set mining, we further compared the expression level of ALDOC between the recurrent tumor site and its counterpart from the same patient (primary site). Of all 20 corresponding patients with primary and following recurrence tumor, we detected the suppression of ALDOC presented in recurrence tumor (*p* = 0.0154, Figure 2D). Kaplan–Meier analyses showed patients with low ALDOC expression significantly correlated with poor overall survival time (*p* = 4.2 × 10^−26^, Figure 2E).

### 2.3. In Silico Predicted Aldolase Family in Glioma Cohorts and Several Benign or Malignant Cell Lines

To characterize aldolase family members in glioma, we also recruited several clinical cohorts to perform the statistical analysis and compare with TCGA cohorts. Our forest plot revealed that expression level of *ALDOC* correlated with better survival in clinical cohorts, including GSE4412, MGH, and GSE4271 (*p* = 2.5 × 10^−2^, *p* = 5.2 × 10^−4^ and *p* = 8.3 × 10^−3^, respectively, Figure 3A). We further summarized aldolase family members in multiple cancer types from the PRECOG website. The data from the PRECOG database showed negative regulating of ALDOC expression in brain-related cancer, including astrocytoma, glioblastoma, glioma, medullobalstoma, meningioma, and neuroblastoma (Figure 3B). Additionally, we also recruited another clinical cohort, CGGA (Chinese Glioma Genome Atlas), and found the consistent trend that expression level of ALDOC correlated with survival curve in clinical patients (Figure 3C). Furthermore, we detected the mRNA level of IDH1 and ALDOC in nine glioma cancer cell lines. However, our data showed that IDH1 and ALDOC did not have strong a correlation (Appendix A). 

### 2.4. ALDOC Expression Associated with IDH1 Mutation and Several Glioblastoma Subtypes

In order to detect the significant correlation between ALDOC and clinical parameters in glioma tumors, we picked up the probes of aldolase family in GSE36245. From the heat map and boxplot, ALDOC expression in IDH1 mutant patients (*n* = 10) is higher than the IDH1 wild-type group (*n* = 29) (*p* = 0.0384, Figure 4A,B). Moreover, our results reveal lower ALDOC expression level in wild-type (*n* = 165) compared with mutant form (*n* = 132) in the CGGA cohort (*p* < 0.001, Figure 4C). Furthermore, we also validated this phenomenon in several clinical cohorts from the GlioVis website (http://gliovis.bioinfo.cnio.es/). Similarly, most of all cohorts presented ALDOC expression in IDH1 mutant patients is higher than IDH1 wild-type groups (Figure 4D). 

### 2.5. ALDOC Expression Level Associated with Prognosis in High-Grade Gliomas

Our results show that ALDOC expression significantly positively correlated with long-term survival in GSE53733 group (Figure 5A,B). However, non-mutated IDH1 expression did not reveal significant prognostic correlation (Appendix A). Besides, we provided evidence that expression level of ALDOC may link to tumor grade. GSE4290 showed significant suppression of ALDOC in glioblastoma patients with non-tumor or low-grade groups (Figure 5C). In addition, compared with control (normal brain tissue, *n* = 8), a little lower ALDOC expression was detected in glioma patients (*n* = 276) (Appendix A). We also proved that higher ALDOC expression in proneural subtypes than in mesenchymal type of glioblastomas was identified from the GSE54077 data set (Appendix A). Through GSE4412 analysis, we found ALDOC expression in grade IV lower than grade III gliomas (Figure 5D,E). Furthermore, ALDOC might be the prognostic factor in high-grade gliomas (Figure 5F). 

### 2.6. Low ALDOC Protein Expression Associated with Poor Prognosis in Human Glioma Tissue

The aldolase family members and clinicopathological parameters are summarized in Table 1 and Table 2 and Appendix A. Of all 64 glioma samples, low ALDOC expression level was observed in 37 (57.8%) cases. In this study, we also examined whether ALDOC expression presented in tumor blood vessels and perinecrotic areas. However, no ALDOC staining was detected in the above two areas (Appendix A). Besides, patients with low ALDOC expression significantly correlated with shorter overall survival time (*p* = 0.006, Figure 6A,B). On the other hand, ALDOA protein expression revealed no statistical significance with the overall survival time in gliomas (*p* = 0.214, Figure 6B). Cox survival analysis was performed to determine their respective prognosis significance. Besides, we focused on glioblastoma patients and results show ALDOC still significantly *p*-value with survival rate in our cohort (*p* = 0.002, Figure 6C). The univariate Cox survival analysis revealed that low ALDOC expression level (hazard ratio [HR] = 0.497; 95% confidence interval [CI] = 0.292–0.846; *p* = 0.010), high grade (hazard ratio [HR] = 3.031; 95% confidence interval [CI] = 1.691–5.433; *p* < 0.001) remained independently prognostic factors for overall survival time (Figure 6D). We also performed several clinical parameters for the glioblastoma cohort to evaluate the biostatical power and reality. Our data also show that age, expression level of AxL, TP53 mutant status, or loss of ATRX could be regarded as independent prognostic factors (Figure 6D and Table 1 and Table 2). Therefore, low ALDOC expression and advanced tumor grade were significantly correlated with unfavorable overall survival. 

### 2.7. ALDOC Involved Proliferation Rate and Metastatic Ability in GBM Cells

To verify our observations from several in silico data sets and clinical cohorts, we further establisished the two-way models, including overexpression and knockdown of ALDOC in GBM cell lines. In this study, we established the ALDOC overexpression model in T-98G and LN-229 cells, and created the ALDOC knockdown model in U-87MG cells. From our results, the enhancement of GBM cell proliferation ability when ALDOC was blocked (Figure 7A,B). On the other hand, we also demonstrated that ALDOC overexpression could reduce the migration/invasion ability of GBM cells (Figure 7C and Appendix A). Moreover, we performed ALDOC-based transcriptomics studies through microarray chips. After normalization and >1.5-fold change cut-off, we selected candidate probes and predicted several canoncial pathways and upstream factors from Ingenuity Pathway Analysis (IPA) software. We observed that TWIST1 was one of the significant transcription factors, activated when ALDOC is suppressed. Therefore, we proposed that ALDOC regulation of migration/invasiveness in GBM cells might be correlated with epethelial-mesenchymal transition (EMT) (Figure 7D). According to all of the above evidences, we concluded that ALDOC expression levels were associated with several phenotypes in GBM and served as an indepedent prognostic factor for GBM patients.

## 3. Discussion

The most common point of mutation of IDH1 is localized to codon 132 and IDH2, located at codon 172 [16]. Of all gliomas, IDH1 revealed a higher mutation rate than IDH2 [16]. The detection of IDH1/2 mutation is predominantly on diffuse astrocytomas, anaplastic astrocytomas, and secondary glioblastomas, but rarely identified on primary glioblastomas [16,17]. In the recent studies, some genetic aberrations are associated with IDH1/2 mutation, including TP53 and ATRX mutation and 1p/19q co-deletion [18]. Suzuki et al. have demonstrated that IDH1-WT gliomas present a poorer prognosis than IDH1-mutant gliomas [19]. Additionally, the status of IDH1/2 could influence the therapeutic response from chemotherapy or radiotherapy [20]. Therefore, the prognosis and optimal therapeutic regimen in primary glioblastomas is difficult to evaluate because of the loss of IDH expression in most cases [21]. In this study, we proved that the aldolase family revealed inverse correlations with survival time in glioblastomas from the TCGA database. Besides, the expression of ALDOC is not only negatively associated with IDH1, but is also related to better prognosis in gliomas. Unlike low-grade glioma (LGG), GBM cancer cell lines do not have IDH1 mutants available for experimental applications. In recent years, IDH1 R132 mutant was been knock-in through CRISPR/Cas9 gene editing [22]. Through established transcriptomics data sets, proteomics, and metabolomics between parental or IDH1 R132 mutant in U-87 cells, combined with our available microarray data sets from ALDOC knockdown models, we expected to find the interaction partners, signaling pathways, or metabolites by ALDOC regulated in only wild-type or R132 mutant of IDH1 in GBM. 

Recent studies have emphasized the relationship between metabolic reprogramming and some oncogenic factors in gliomas [23]. The characters of low oxygen level in glioma cells could enhance some hypoxia-inducible factors, including mainly Hypoxia-inducible factor 1-alpha (HIF-1α). After then, the role HIF-1α activated 5’-AMPK-activated kinase (AMPK) signaling and upregulated CreB expression, which stimulated glycolytic enzymes to produce energy for glioma cells [24]. Aldolase has been recognized one of the HIF-1α downstream factors. They have hypoxia-response elements (HRE) in the promoter region and form a positive feedback loop in tumorigenesis [25]. Moreover, HIF-1α targets Vascular Endothelial Growth Factor (VEGF) and accelerates angiogenesis for tumorigenesis and metastasis. In addition, Zhao et al. demonstrated that mutated IDH1 could keep the stability of HIF-1α existence, which proceeded glycolytic pathway and resulted in gliomagenesis [26]. Otherwise, Caspi et al. also proved that ALDOC played a role of positive regulator of Wnt signaling pathway and induced tumorigenesis [14]. ALDOA has been mentioned for its lower Km and higher enzymatic activity, compared with ALDOB and ALDOC, to process the glycolysis and Warburg effect in tumorigenesis. On the other hand, ALDOB plays a key role in fructolysis with KHK enzyme. For ALDOC, only limited evidence implies that ALDOC overexpression is related to brain development and injury. We proposed that ALDOC plays an important role in regulating metabolic events and may reveal stronger binding affinity with interaction common partners than ALDOA and ALDOB. Therefore, we utilized proteomics/phosphoproteomics to claim whether ALDOC is a tumor suppressor in GBM. Furthermore, we found that ZFP36 was down-regulated in the ALDOC knockdown data sets. In recent studies, loss function of ZFP36 could impair GBM cell viability and invasiveness [27]. In this study, we evaluated the association of this axis and gliomagenesis in future experiments. From our results, higher ALDOC mRNA expression in proneural subtype than in mesenchymal subtype of glioblastomas might imply the interaction of ALDOC with various oncogenic mutations, and possibly influence overall survival time. Moreover, more evidence is needed to verify the detailed mechanism. In the future, a newly-developed, targeted activator of ALDOC might be a potential effector to slow down tumor progression after verification.

From aforementioned results, we did not observe a significant correlation between IDH1 expression levels and ALDOC in the glioblastoma cell lines panel. The possible reason might be related to the characteristics of heterogeneity of glioblastoma. Furthermore, the glioblastoma cell lines lack the IDH1 R132 mutant form for the experiments predicted (Appendix A). Therefore, we will try to create the glioblastoma with IDH1 knockout stable cells by CRISPR strategy and further knock-in several mutant forms of IDH1, including R132H, R132Q, and S92A, through virus infection. 

In past literatures, ALDOB is an abundant protein in the liver, stomach, and intestines. Besides, ALDOB is a key factor of fructolysis, leading to hereditary fructose intolerance (HFI). Recently, it has been mentioned that fructose metabolic dysfunction might induce several non-neoplastic diseases and cancers [28]. Not only lymphoreticular tissue and placenta, but also brain and reproductive tissues are possible additional organs of fructose metabolism [29]. Therefore, we detected ALDOB overexpression in GBM patients, which when correlated with poor survival (Figure 6C) might imply that the metabolism reprogramming in GBM depends on ALDOB expression. Furthermore, we evaluated the dynamic changes, expression level and enzyme activity of all aldolase family members in future investigations.

The observation of ALDOC, mainly in cerebellar Purkinje cells and retina, is responsible for neural cells’ and fibers’ repairment after brain injury [30]. A recent study proved that the expression of ALDOC in Parkinson’s disease was often accompanied by some atypical clinical symptoms, but rarely associated with disease progression [31]. In this study, we successfully detected the expression of ALDOA, ALDOB, and ALDOC and associated prognosis in various malignancies based on the TCGA database. Furthermore, our results also imply that ALDOC mRNA and protein expression are inversely correlated with non-mutated IDH1 expression and play a good prognostic role in gliomas. In the future, although the detailed mechanism is still unclear, ALDOC might be viewed as a potentially proper target to predict tumor behavior and patient prognosis in glioblastomas. 

## 4. Materials and Methods

### 4.1. In Silico Study

The clinical information and genomic matrix file of The Cancer Genome Atlas (TCGA) database were download from the USCS Xena browser website (https://xenabrowser.net/heatmap/). All Gene Expression Omnibus Series (GSE) serious data sets downloaded from Gene Express Omnibus (GEO) website, normalized, and analyzed through Genspring software (Version 13.1.1, Agilent, Santa Clara, CA, USA). The entire data we downloaded included clinical parameters and expression level of target genes in glioma patients from Xena browser. This website applied the microarray or next-generation sequencing of each probe after normalization. For high expressions, they set the median to be higher than the high expression, and vice versa. In addition, we removed several clinical cases that lacked the corresponding parameters. We further divided them into two groups: Low-grade glioma and glioblastoma, and we observed that ALDOC was inhibited in patients with glioblastoma. The statistical analyses were performed using SPSS 17.0 software (SPSS, Inc., Chicago, IL, USA). The differences between two groups were analyzed using a paired *t*-test or a Mann-Whitney U test. All *p*-values of less than 0.05 were considered statistically significant.

### 4.2. Case Selection

In total, 64 patients diagnosed with various grades of gliomas at the Tri-Service General Hospital in Taiwan from 1991 to 2005 were included in this study. All the included cases were reclassified as 1 pilocytic astrocytoma, 3 diffuse astrocytomas with IDH mutant, 7 diffuse astrocytomas with IDH wildtype, 3 anaplastic astrocytomas with IDH mutant, 6 anaplastic astrocytomas with IDH wildtype, 3 glioblastomas with IDH mutant, 24 glioblastomas with IDH wildtype, 11 diffuse midline gliomas, H3 K27M-mutant, 2 oligodendrogliomas, NOS, and 4 anaplastic oligodendrogliomas, NOS by immunohistochemical stains. No patient had ever received preoperative chemotherapy or radiation therapy. Clinical information and pathology data were collected via a retrospective review of patient medical records. All cases were diagnosed according to the newest version of World Health Organization (WHO) classification^1^. Follow-up data were available in all cases, and the longest clinical follow-up time was 60 months. Written informed consent for the biological studies was obtained from each patient involved in the study, and the study was approved by the Institutional Review Board, Tri-Service General Hospital (numbered as 098-05-295).

### 4.3. Tissue Microarray Construction and Immunohistochemical Staining

Representative 1-mm-diameter cores from each tumor taken from the formalin-fixed paraffin embedded tissue were selected by morphology typical of the diagnosis. The histopathologic diagnoses of all samples were reviewed and confirmed by a pathologist via hematoxylin and eosin-stained slides. IHC staining was performed on serial 5-μm-thick tissue sections cut from the tissue microarray (TMA) using an automated immunostainer (Ventana Discovery XT autostainer, Ventana Medical Systems, Tucson, AZ, USA). Sections were first dewaxed in a 60 °C oven, deparaffinized in xylene, and rehydrated in graded alcohol. Antigens were retrieved by heat-induced antigen retrieval for 30 min with pH 8.0 TRIS-EDTA buffer. Slides were stained with a polyclonal rabbit anti-human ALDOA antibody (1:500, ab71433, Abcam, Cambridge, UK), ALDOB (1:50. ab133333, Abcam, Cambridge, UK), and ALDOC (1:400. Cat.T0906, Abcam (Epitomics), Cambridge, UK). 

### 4.4. TMA Immunohistochemistry Interpretation

The IHC staining assessment was independently conducted by a pathologist blinded to patient outcome. For aldolase family members’ IHC staining, only cytoplasmic IHC expressions of tumor cells in the cores were evaluated. We scored tumor of ALDOC expression using intensity scores of 0, 1, or 2 in tissue microarray. We also evaluate percentage scores from 0~100%. Finally, we measured the total IHC score by intensity × percentage, then 50% cut-off for high expression and low expression groups. Both the immunoreactivity intensity and percentage were recorded. Immunostaining interpretation was performed as described in a previous study. More than 5% of tumor cell staining was considered as a positive result; no cytoplasmic staining or cytoplasmic staining in <5% of tumor cell was defined as score 0. A score of 2+ was defined as high expression and considered to have retained gene expression. Scores of 0 and 1+ were defined as low expression and indicated as loss of expression of candidate genes. 

### 4.5. Cell Lines and Cell Culture Conditions

GBM cell lines T-98G and U-87MG were maintained in MEM supplemented with 10% FBS and 1% Penicillin-Streptomycin-Glutamine (PSG) (Invitrogen). LN-229 was maintained in DMEM medium supplemented with 10% FBS and 1% PSG. All GBM cell lines were acquired from the American Type Culture Collection (ATCC).

### 4.6. Microarray Analysis

Total RNA from U-87MG cells treated with or without shALDOC were extracted with TRIzol RNA extraction kit (Invitrogen). cDNA synthesis from total RNA and microarrays hybridization/scanning were performed with Affymetrix GeneChip products (HG-U133A) by GRC Microarray Core Facility (Academia Sinica, Taiwan). Data files (CEL) were converted into probe set values (log2) by RMA normalization using GeneSpring (Agilent). 

### 4.7. Statistical Analysis

The non-parametric Mann-Whitney U-test was used to analyze the statistical significance of results from three independent experiments. Statistical analyses were performed using SPSS (Statistical Package for the Social Sciences) 17.0 software (SPSS, Chicago, IL, USA). The association between clinicopathological categorical variables and the ALDOC IHC expression levels were analyzed by Pearson’s chi-square test. Estimates of the survival rates were calculated using the Kaplan–Meier formula and compared using the log-rank test. Follow-up time was censored if the patient was lost during follow-up. Univariate and multivariate analyses were performed using Cox proportional hazards regression analysis with and without an adjustment for ALDOC IHC expression level, grade. For all analyses, a *p*-value of <0.05 was considered significant.

## 5. Conclusions 

In this study, we integrated The Cancer Genome Atlas (TCGA) database and several microarray data sets from the Gene Expression Omnibus (GEO) database. Here, we identified glycolysis enzyme aldolase C (ALDOC) correlated with several genetic events, including IDH mutant, MGMT methylation status, chromosome (1p,7,10 and 19q), or several clinicopathological parameters. We further investigated ALDOC correlated with survival curve and tumor grade in brain tumor patients. Moreover, the performance of ALDOC was significantly decreased in tumor parts compared with normal adjacent tissues in RNA and protein levels. These findings prove that loss of ALDOC expression is an independent factor for predicting poor prognosis and has a stronger prognostic ability than ALDOA/ALDOB in glioblastoma patients.

## Figures and Tables

**Figure 1 cancers-11-01238-f001:**
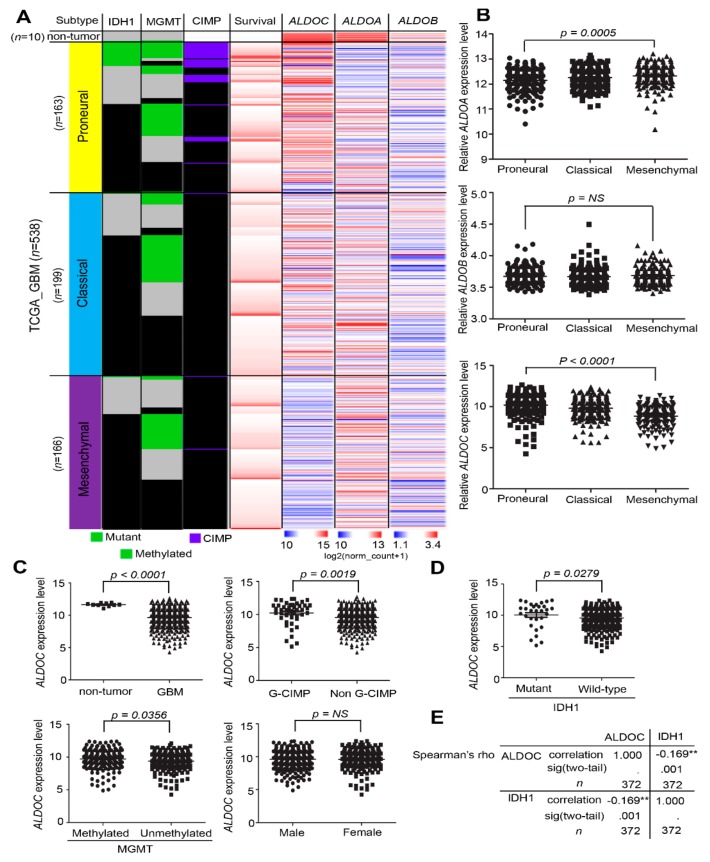
ALDOC mRNA expressed in various subtypes of glioblastomas and correlated with several parameters. (**A**) A heat map showing the endogenous mRNA expression level of the aldolase family members ALDOA, ALDOB, and ALDOC, subtypes, IDH1 mutant status, MGMT methylated status, CIMP status, and survival time in TCGA_Glioblastoma patients (*n* = 538). (**B**) A boxplot showing the distribution of each aldolase family member’s expression in clinical patients according to the molecular subtypes in glioblastoma (ALDOA: *p* = 0.0005, ALDOB: *p* = non-significant (NS), ALDOC: *p* < 0.0001). (**C**) A boxplot showing the distribution of ALDOC expression in clinical patients according to histology, CIMP status, MGMT methylated status, and sex in the glioblastomas (*n* = 538, *p* < 0.0001, *p* = 0.019, *p* = 0.0356, and *p* = NS, respectively). (**D**) A boxplot showing the distribution of ALDOC expression in clinical patients according to the IDH1 mutant status in the glioblastomas (*n* = 538, *p* = 0.0279). (**E**) The correlation between the ALDOC mRNA level and the mutation event of IDH1 in glioblastoma patients from the TCGA cohort (spearman’s rho = ~0.169, *p* = 0.001). The significance of the differences in B was analyzed using the one-way ANOVA formula. The significance of C and D were analyzed using a Student’s *t*-test. All cut-off values were set before analysis, and all tests were two-tailed. (** *p* < 0.05.)

**Figure 2 cancers-11-01238-f002:**
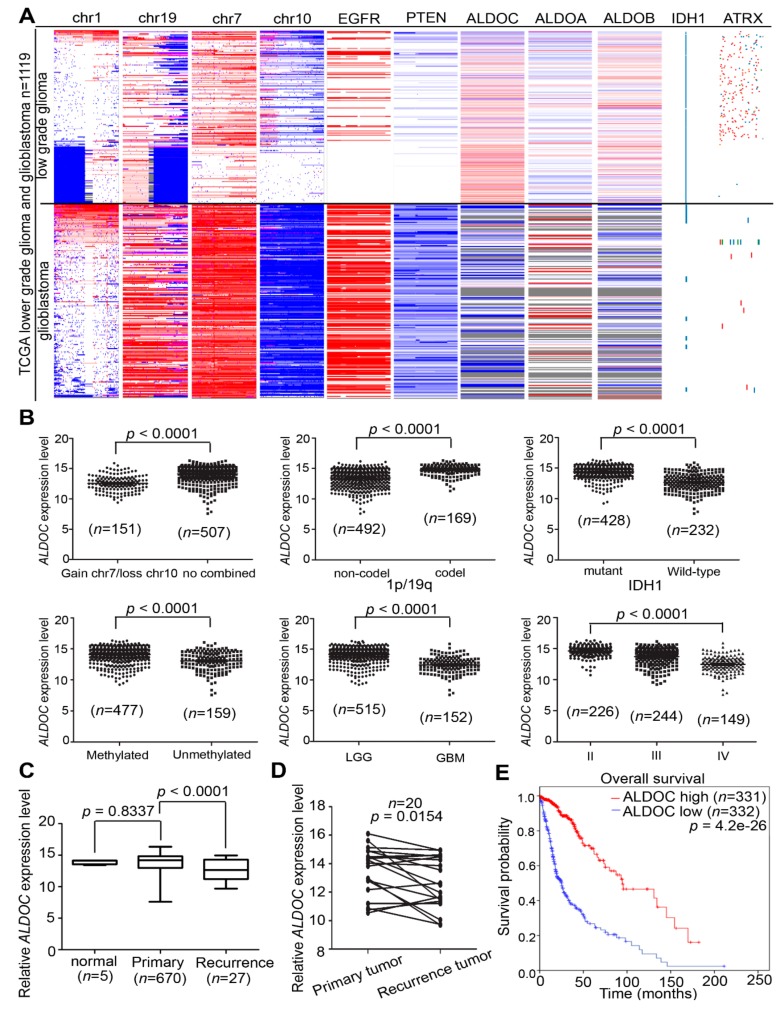
Downregulation of ALDOC mRNA expression in recurrence tumor associated with poor survival. (**A**) A heat map showing the endogenous mRNA expression level of the Aldolase family members ALDOA, ALDOB, and ALDOC in TCGA_ GBMLGG (*n* = 1119). (**B**) a boxplot showing the distribution of ALDOC expression in clinical patients according to the chromosome (1p/19q, 7 and 10) events and grade in TCGA_GBMLGG (*n* = 1119). (**C**) A boxplot showing the distribution of ALDOC expression in clinical patients according to the normal, primary, or recurrence tumor in glioblastoma (*n* = 702, *p* = 0.8337 and *p* < 0.001, respectively). (**D**) Normal adjacent tissue/tumor (N/T) pairs showing the expression of ALDOC in clinical patients according recurrence tumor or not in glioblastoma (*n* = 20, *p* = 0.0154). (**E**) Kaplan–Meier analysis of ALDOC expression at concurrently low or high levels, with the endpoint of overall survival probability from the Kaplan–Meier Plotter database in glioblastoma patients (*p* = 4.2 × 10^−26^). The significance of the differences in B, C, and D was analyzed using a Student’s *t*-test. All cut-off values were set before analysis, and all tests were two-tailed.

**Figure 3 cancers-11-01238-f003:**
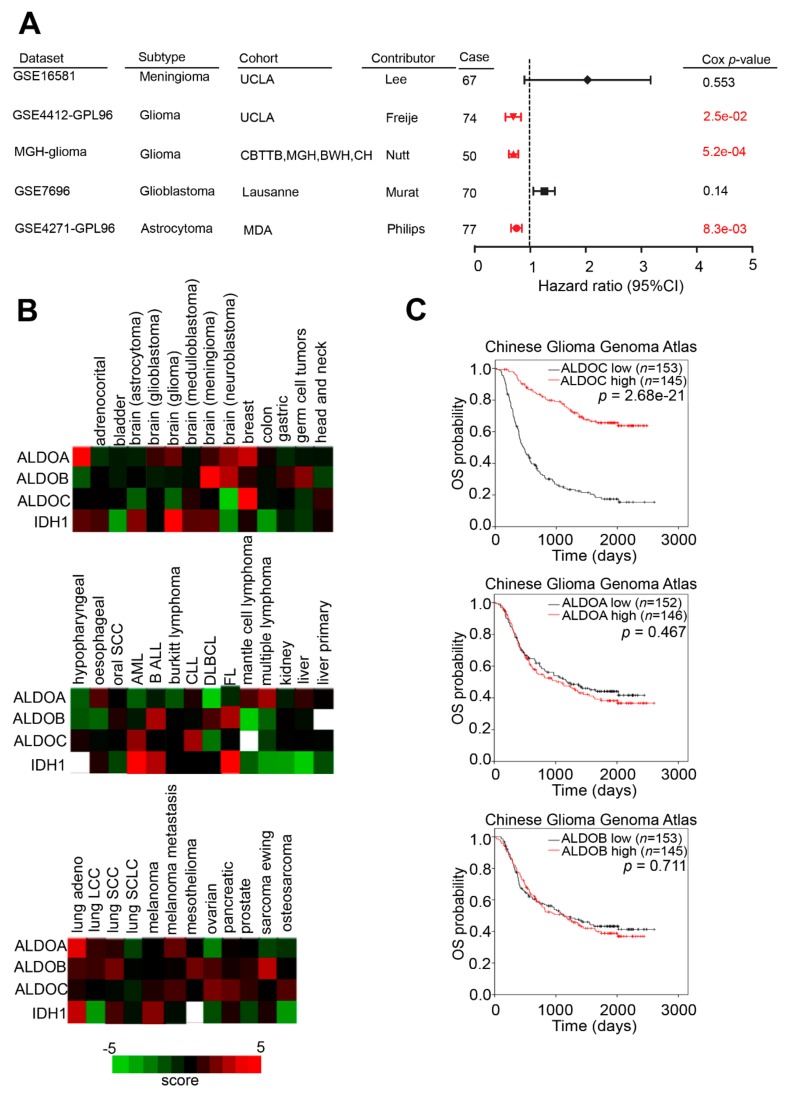
ALDOC mRNA expression associated with clinical features. (**A**) A forest plot showing hazard ratio and 95% confidence interval (CI) by ALDOC expression level in several clinical cohorts. (**B**) A heat map showing the endogenous mRNA expression level of IDH1 and the aldolase family members ALDOA, ALDOB, and ALDOC in various cancer subtypes from the PRECOG website (*n* = 720). (**C**) Kaplan–Meier analysis of ALDOA, ALDOB, and ALDOC expression at concurrently low or high levels, with the endpoint of overall survival probability from the Chinese Glioma Genome Atlas in glioblastoma patients (*p* = 711, *p* = 0.467, and *p* = 2.7 × 10^−21^, respectively).

**Figure 4 cancers-11-01238-f004:**
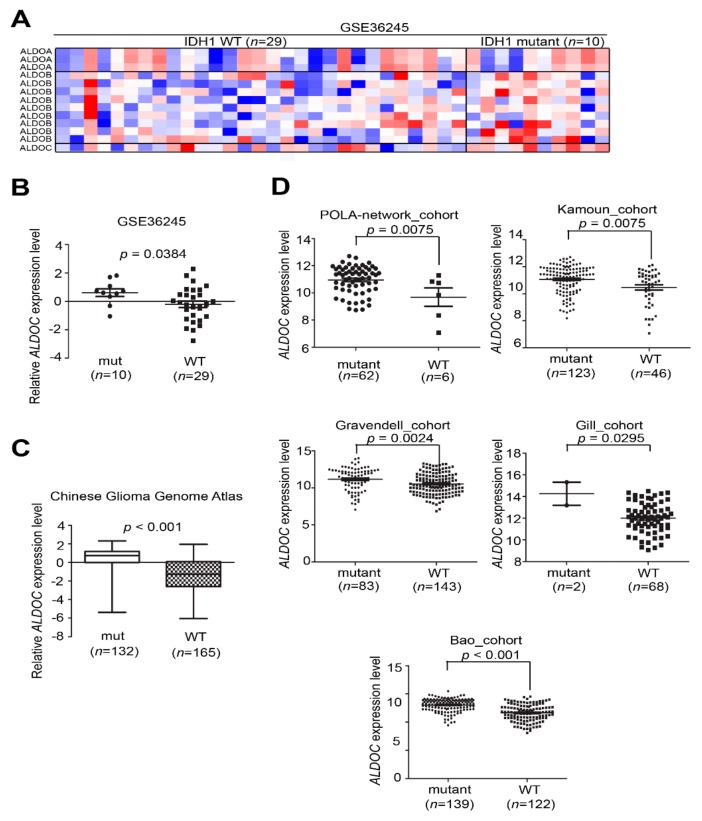
ALDOC mRNA expression associated with IDH1 mutation event in silico data sets. (**A**) A heat map showing the endogenous mRNA expression level of aldolase family members ALDOA, ALDOB, and ALDOC in GSE36245 (*n* = 38). (**B**) A boxplot showing the distribution of ALDOC expression in clinical patients according to the IDH1 mutant event in the GSE36245 cohort (*p* = 0.0384). (**C**) A boxplot showed the distribution of ALDOC expression in clinical patients according to the IDH1 mutant event in the CGGA cohort (*p* < 0.001). (**D**) A boxplot showing the distribution of ALDOC expression in clinical patients according to the IDH1 mutant event in several cohorts (POLA-network: *p* = 0.0075, Kamoun: *p* = 0.0075, Gravendell: *p* = 0.0024, Gill: *p* = 0.0295, and Bao: *p* < 0.001, respectively). The significance of the differences in B, C, and D was analyzed using a Student’s *t*-test. All cut-off values were set before analysis, and all tests were two-tailed.

**Figure 5 cancers-11-01238-f005:**
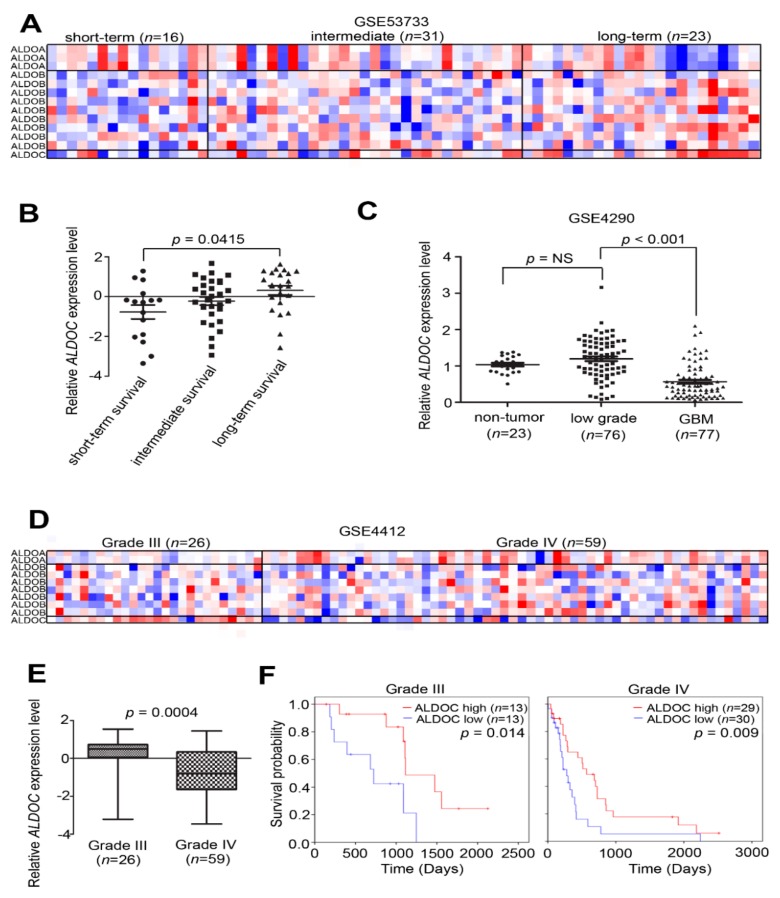
ALDOC mRNA expression correlated with survival curve and tumor grade. (**A**) A heat map showing the endogenous mRNA expression level of aldolase family members ALDOA, ALDOB, and ALDOC in GSE53733 (*n* = 70). Short-term survival: <12 months, long-term survival: >36 months, and 12~36 months as intermediate. (**B**) A boxplot showing the distribution of ALDOC expression in clinical patients according to the survival rate in the GSE53733 cohort. (**C**) A boxplot showing the distribution of ALDOC expression in clinical patients according to the histology in the GSE4290 cohort. (**D**) A heat map showing the endogenous mRNA expression level of aldolase family members ALDOA, ALDOB and ALDOC in GSE4412 (*n* = 84). (**E**) A boxplot showing the distribution of ALDOC expression in clinical patients according to the tumor grade in the GSE4412 cohort (*p* = 0.0004). (**F**) Kaplan–Meier analysis of ALDOC expression at concurrently low or high levels, with the endpoint of overall survival probability by grade III or grade IV from the Kaplan–Meier Plotter database in glioblastoma patients, respectively. The significance of the differences in E was analyzed using a Student’s *t*-test. The significance of the differences in B and C were analyzed using the one-way ANOVA formula. All cut-off values were set before analysis, and all tests were two-tailed.

**Figure 6 cancers-11-01238-f006:**
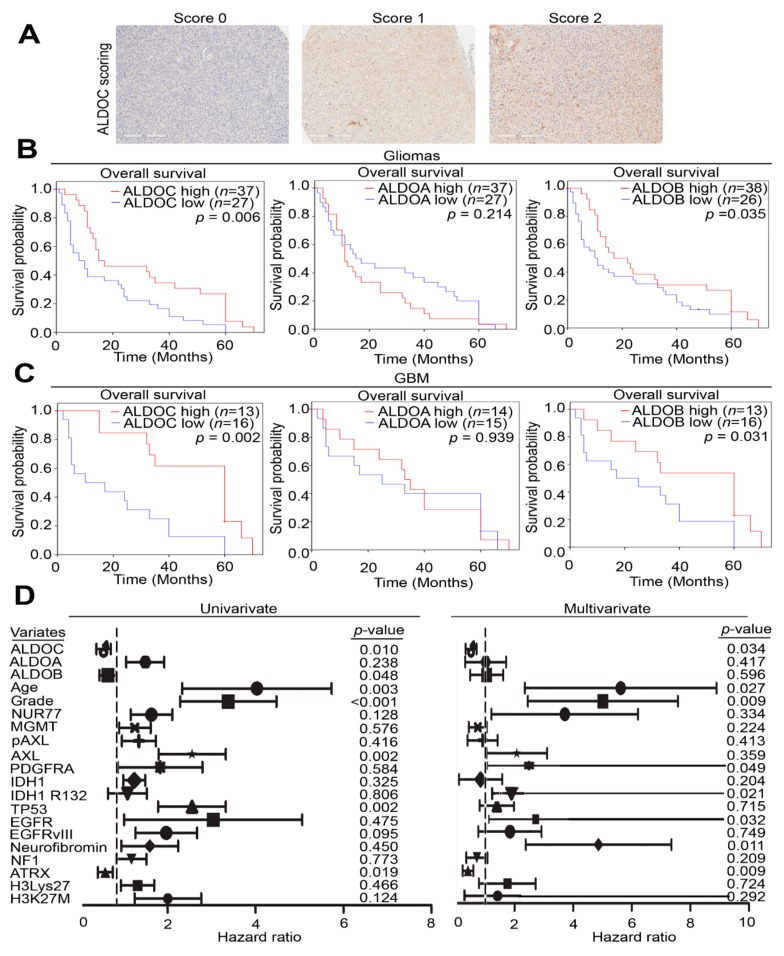
ALDOC protein level as independent prognostic factor in glioblastoma patients. (**A**) The scores (0–2) indicate ALDOC protein level by immunohistochemistry in representative brain tumor tissues. (**B**) Kaplan–Meier analysis of ALDOA, ALDOB, and ALDOC alone at concurrently low or high levels as determined by IHC staining at the endpoint of overall survival probability and disease-free survival probability in brain cancer patients, respectively. (**C**) Kaplan–Meier analysis of ALDOA, ALDOB, and ALDOC alone at concurrently low or high levels as determined by IHC staining at the endpoint of overall survival probability and disease-free survival probability in glioblastoma patients, respectively. (**D**) Multivariate analysis and univariate analysis of ALDOA/ALDOB/ALDOC and clinical parameters in the clinical cohort. The significance of the differences in B and C was analyzed using a Cox progression. All cut-off values were set before analysis, and all tests were two-tailed. informed consent.

**Figure 7 cancers-11-01238-f007:**
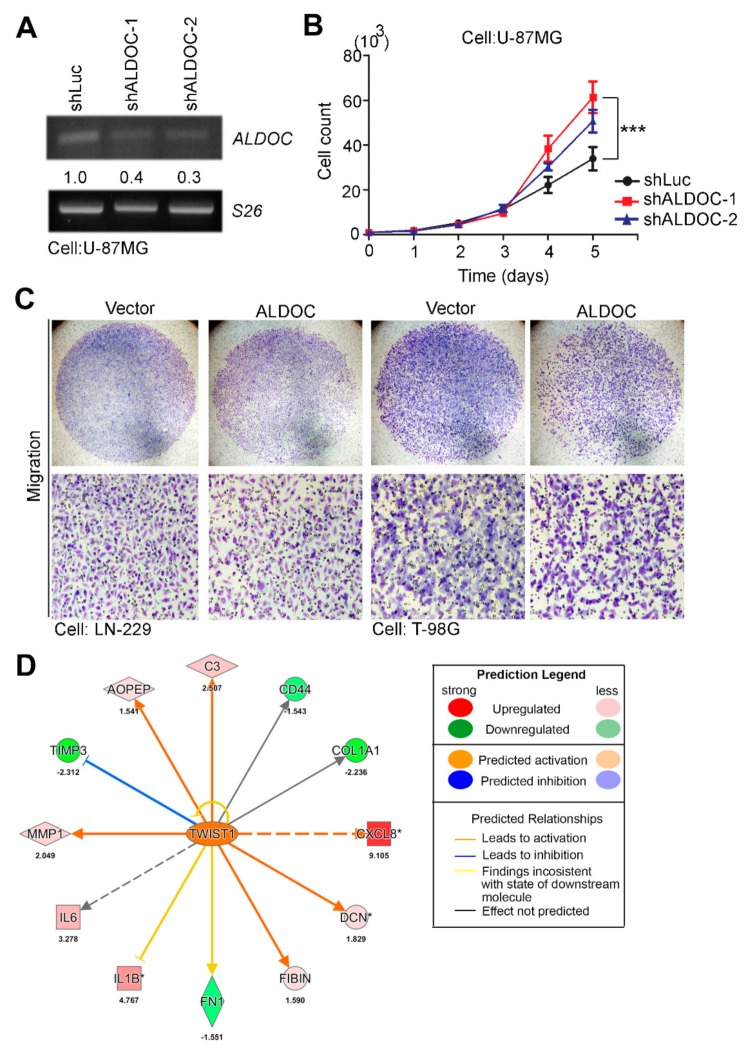
ALDOC is involved in proliferation rate and metastatic ability of glioblastoma cells. (**A**) RT-PCR analysis of ALDOC mRNA expression of the ALDOC knockdown models in U-87MG cells. (**B**) Trypan blue assay to measure cell viabilities of the ALDOC knockdown models in U-87MG cells. (**C**) Representative Giemsa staining to estimate the migration abilities of LN-229 and T-98G cells with forced expression of the vector control or exogenous ALDOC gene, respectively. (**D**) The network was predicted using the common signature that was overlaid with microarray data from U-87MG shALDOC cells with a 1.5-fold change cut-off in the IPA database. (*** p < 0.01.)

**Table 1 cancers-11-01238-t001:** Statistical analyses for ALDOC and clinical parameters expression in brain tumors.

		Univariate Analysis		Multivariate Analysis	
Variable	Total	OR (95% CI)	*p*-value	OR (95% CI)	*p*-value
Sex					
Male	40	0.912–2.648	0.105	1.015–4.429	0.045 *
Female	24				
Age					
<50	12	1.517–7.265	0.003 *	1.158–11.988	0.027 *
≥50	52				
IDH1 R132H					
Negative	9	0.429–1.929	0.803	1.55–230.756	0.021 *
Positive	55				
ATRX					
Preserve	26	0.319–0.902	0.019 *	0.148–0.763	0.009 *
Loss of expression	38				
H3K27M					
Negative	10	0.861–3.441	0.124	0.101–2048.871	0.292
Positive	54				
MGMT					
Preserved	30	0.696–1.921	0.576	0.292–1.334	0.224
Loss of expression	34				
EGFR					
Negative	2	0.403–7.026	0.475	1.229–102.588	0.032 *
Positive	62				
EGFRvIII					
Negative	12	0.908–3.290	0.095	0.370–3.952	0.749
Positive	52				
P53					
Negative	33	1.353–3.990	0.002 *	0.513–2.516	0.715
Overexpression	31				
Neurofilament					
Negative	9	0.631–2.822	0.450	1.343–9.670	0.011 *
Positive	55				
NF1					
Negative	30	0.642–1.814	0.773	0.209–1.412	0.209
Positive	34				
AxL					
Negative	32	1.360–3.988	0.002 *	0.601–4.074	0.359
Positive	32				
p-AxL					
Negative	30	0.740–2.069	0.416	0.208–1.909	0.413
Positive	34				
NUR77					
Negative	33	0.891–2.514	0.128	0.481–8.632	0.334
Positive	31				
H3Lys27					
Preserved	30	0.725–2.020	0.466	0.408–3.641	0.724
Loss of expression	34				
PDGFRA					
Negative	4	0.479–3.689	0.584	1.005–49.672	0.049 *
Positive	60				

*p*-value < 0.05 is considered significant (*).

**Table 2 cancers-11-01238-t002:** Multivariate analyses for various clinical parameters expression in brain tumor.

	Multivariate Analysis		
Variable	Hazard Ratio	95% Confidence Interval	*p*-Value
Gender (male/female)	2.121	1.015–4.429	0.045 *
Age (<50/≥50)	3.726	1.158–11.988	0.027 *
IDH1 R132H (Negative/Positive)	18.911	1.55–230.756	0.021 *
ATRX (Preserve/Loss)	0.336	0.148–0.763	0.009 *
H3 K27M (Negative/Positive)	14.409	0.101–2048.871	0.292
MGMT (Preserve/Loss)	0.624	0.292–1.334	0.224
EGFR (Negative/Positive)	11.231	1.229–102.588	0.032 *
EGFRvIII (Negative/Positive)	1.213	0.370–3.952	0.749
P53 (Negative/Overexpression)	1.156	0.513–2.516	0.715
Neurofilament (Negative/Positive)	3.603	1.343–9.670	0.011 *
NF1 (Negative/Positive)	0.540	0.209–1.412	0.209
AxL (Negative/Positive)	1.564	0.601–4.074	0.359
p-AxL (Negative/Positive)	0.629	0.208–1.909	0.413
NUR77 (Negative/Positive)	2.037	0.481–8.632	0.334
H3 K27me3 (Preserve/Loss)	1.218	0.408–3.641	0.724
PDGFRA (Negative/Positive)	7.067	1.005–49.672	0.049 *

*p*-value < 0.05 is considered significant (*).

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
