# Peer review of "Enrichment of Aldolase C Correlates with Low Non-Mutated IDH1 Expression and Predicts a Favorable Prognosis in Glioblastomas"

_cancers, 2019, doi:10.3390/cancers11091238_

Round 1

Reviewer 1 Report

In this manuscript, authors describe the potential role of aldolase family in glioma development. They focused on ALDOC which seems to be a good prognosis of patient survival.

The results are well presented and clear.

However, some major points need to be clarified and some extra experiments have to be performed before accepting this manuscript.

Major concerns :

- in Figure 6A : the IHC stainings are not easy to analyze. The authors may magnify the images for showing where ALDOC is expressed. Is ALDOC expressed in all tumor cells ? In tumor blood vessels ? Perinecrotic areas ?

- in Figure 6C, ALDOB is shown to be a good prognosis marker. Can the authors discuss this result in the manuscript ? Some explanations on aldolase specificity would be appreciated.

- Some in vitro experiments would be appreciated : can the authors overexpress ALDOC in glioblastoma cell lines in order to observe the consequences on cell viability, migration, invasion ?

Author Response

Reviewer 1:

In this manuscript, authors describe the potential role of aldolase family in glioma development. They focused on ALDOC which seems to be a good prognosis of patient survival. 

The results are well presented and clear. 

However, some major points need to be clarified and some extra experiments have to be performed before accepting this manuscript. 

Major concerns : 

in Figure 6A : the IHC stainings are not easy to analyze. The authors may magnify the images for showing where ALDOC is expressed. Is ALDOC expressed in all tumor cells ? In tumor blood vessels ? Perinecrotic areas ? 

Answer: We have magnified the images and added the arrows to indicate where did ALDOC stain. From our results, ALDOC expressed in some degree of glioma cytoplasm, but not in all tumors. (please see the figure 6A in revised manuscript)

In this study, we also have checked if ALDOC expression presented in tumor blood vessels and perinecrotic areas. However, no ALDOC staining was detected in above 2 areas. Some related figures were listed as follows:

(please also see the supplement figure 5 in revised manuscript)

in Figure 6C, ALDOB is shown to be a good prognosis marker. Can the authors discuss this result in the manuscript ? Some explanations on aldolase specificity would be appreciated.

Answer: In past literatures, ALDOB is an abundant protein in liver, stomach and intestine. Besides, ALDOB is a key factor of fructolysis, leading to hereditary fructose intolerance (HFI). Recently, it has been mentioned that fructose metabolic dysfunction might induce several non-neoplastic diseases and cancers (Trends Endocrinol Metab. 2018 Aug;29(8):549-559.). Not only lymphoreticular tissue, placenta but also brain and reproductive tissues are possible additional organs of fructose metabolism (Brain Res. 2017 Feb 15;1657:312-322.). Therefore, we detected ALDOB overexpression in GBM patients and correlated with poor survival (Figure 6C), might imply the metabolism reprogramming in GBM depend on ALDOB expression. Furthermore, we evaluated the dynamic changes, expression level and enzyme activity of all aldolase family members in future investigations. (please see line 227-235 of page 5 in revised manuscript.)

Some in vitro experiments would be appreciated: can the authors overexpress ALDOC in glioblastoma cell lines in order to observe the consequences on cell viability, migration, invasion ?

Answer: We established ALDOC knockdown stable cells by shRNA. We further evaluated migration/invasive ability in LN-229 and T-98G cells through Boyden’s chamber. Our data shows that when ALDOC is overexpressed, the migration/invasive ability will increase.

Representative images of migration/invasion in the LN-229 and T-98G ALDOC overexpression model, respectively.

Reviewer 2 Report

The authors in this study has shown that the loss of ALDOC expression is a prognostic factor in glioblastoma patients. Major comments The article seems to be of educational importance. More elaborate and recent literature citations should be included in the introduction section, that will provide a better understanding of the significance of the manuscript. In addition, authors should emphasize on why this study is novel in few sentences compared to other published studies in reference to ALDOC expression. The study is well-designed and appropriate statistical tools are utilized in the study. My biggest concern with the study is that the "n' for controls are way less compared to experimental group. Also, the analysis should be conducted separately for males and females, as previous studies have shown that there are sex differences in glioblastoma. Minor Comments: The manuscript needs improvement in grammar and syntax. It should be carefully edited to facilitate coherency.

Author Response

Reviewer 2:

The authors in this study has shown that the loss of ALDOC expression is a prognostic factor in glioblastoma patients. Major comments The article seems to be of educational importance. More elaborate and recent literature citations should be included in the introduction section, that will provide a better understanding of the significance of the manuscript. In addition, authors should emphasize on why this study is novel in few sentences compared to other published studies in reference to ALDOC expression. The study is well-designed and appropriate statistical tools are utilized in the study. My biggest concern with the study is that the "n' for controls are way less compared to experimental group. Also, the analysis should be conducted separately for males and females, as previous studies have shown that there are sex differences in glioblastoma. Minor Comments: The manuscript needs improvement in grammar and syntax. It should be carefully edited to facilitate coherency. 

Answer: We are appreciated the critical comments and kind reminder from reviewer #2. We are collecting patient specimens to expand the available clinical cohort including tumor parts and its adjacent normal tissue. In the current absence of clinical cases, we have conducted various in silico datasets to validate our hypotheses. Perhaps the proportion of male/female in our cohort is unbalanced (n=40/24, respectively). We observed gender as the target revealed a boarding line in the multivariate analysis (p=0.045). However, we need reassess that gender may be not an independent prognostic factor in our cohort. If necessary, we will send the manuscript to the language editing service.

Reviewer 3 Report

Chang et al., analyzed the correlation between Aldolase C (ALDOC) and glioblastoma (GBM) using several GBM datasets. They found that higher ALDOC mRNA in normal brain tissue compared to GBM, and higher ALDOC in LGG compared to GBM. Moreover, they identified that ALDOC negatively correlated with non-mutated IDH1 expressions, as well as positively associated with patient survival. Overall, these findings are very interesting.

I have the following recommendations for improvement:

1. The key findings of this manuscript were based on bioinformatic analysis. It will be very important to validate the key findings using cell culture and mouse models; For example, the author may want to study whether knockdown and inhibition of ALDOC can affect glioma cell proliferation and survival in vitro, and tumor growth and progression in vivo.

2. As the authors mentioned ALDOC involves in metabolism and glycolysis. Glycolysis is very important for GBM progression. It is unclear why ALDOC can function as a tumor suppressor in GBM.

3. In Fig S2, the authors showed that there is no correlation between IDH1 and ALDOC in 9 glioma cell lines. The authors may want to investigate whether ALDOC expression is related to IDH1 mutation status in a panel of glioma cell lines. On the other hand, they author may want to use CRISPR KO or shRNA KD approach to confirm these results.

4.  How to explain the gender difference?

5. It is unclear why ALDOC is enriched in proneural subtype compared to mesenchymal subtype. Does ALDOC can inhibit GBM mesenchymal differentiation?

Author Response

Reviewer 3:

Chang et al., analyzed the correlation between Aldolase C (ALDOC) and glioblastoma (GBM) using several GBM datasets. They found that higher ALDOC mRNA in normal brain tissue compared to GBM, and higher ALDOC in LGG compared to GBM. Moreover, they identified that ALDOC negatively correlated with non-mutated IDH1 expressions, as well as positively associated with patient survival. Overall, these findings are very interesting.

I have the following recommendations for improvement:

The key findings of this manuscript were based on bioinformatic analysis. It will be very important to validate the key findings using cell culture and mouse models; For example, the author may want to study whether knockdown and inhibition of ALDOC can affect glioma cell proliferation and survival in vitro, and tumor growth and progression in vivo.

Answer: Thank you for the kind suggestion. We have established ALDOC shRNA knockdown clones in low endogenous expression cell line U-87MG. We further measure the cell viability between ALDOC knockdown groups with vector control. Our results performed that inhibition of ALDOC can increased cell viability in vitro. We will evaluate the tumorigenecity in vivo through subcutaneous inject in future.

(Left) Expression level of ALDOC in U-87 cells with or without shALDOC clones. (Right)Cell proliferation ability in U-87 cells with or without shALDOC clones

As the authors mentioned ALDOC involves in metabolism and glycolysis. Glycolysis is very important for GBM progression. It is unclear why ALDOC can function as a tumor suppressor in GBM.

Answer: ALDOA has been mentioned its low Km and high enzymatic activity than ALDOB and ALDOC to process the glycolysis and Warburg effect in tumorigenesis. On the other hands, ALDOB play the key role in fructolysis with KHK enzyme. For ALDOC, it is only rare evidence to point out the ALDOC overexpress in brain development and injury. We proposed that ALDOC play the keeper to control metabolic events and may strongly binding affinity with interaction common partners than ALDOA and ALDOB. Therefore, we will utilize proteomics/phosphoproteomics to claim whether ALDOC is a tumor suppressor in GBM. Furthermore, we found that ZFP36 was suppressed in the ALDOC knockdown datasets. ZFP36 loses function to impair glioblastoma cell viability and invasiveness (Cell Cycle. 2012 May 15;11(10):1977-87.). We will also prove this axis in future experiments. (please see line 205-214 of page 5 in revised manuscript.)

The potential upstream regulators were been predicted by Ingenuity Pathway Analysis (IPA) software based on microarray from U87 shALDOC cells with a 1.5-fold change cut-off compare with vector control.

In Fig S2, the authors showed that there is no correlation between IDH1 and ALDOC in 9 glioma cell lines. The authors may want to investigate whether ALDOC expression is related to IDH1 mutation status in a panel of glioma cell lines. On the other hand, they author may want to use CRISPR KO or shRNA KD approach to confirm these results.

Answer: In figure S2, we only observed there is no correlation between RNA level of IDH1 and ALDOC. However, there is currently no evidences that ALDOC and IDH1 are regulated at the protein level or metabolic events. Unlike low grade glioma (LGG), GBM cancer cell lines do not have IDH1 mutants available for experimental applications. Recent years, IDH1 R132 mutant was been knock-in through CRISPR/Cas9 gene editing (J Biol Chem. 2017 May 12;292(19):7971-7983. ). Through establish transcriptomics datasets, proteomics and metabolomics between parental or IDH1 R132 mutant in U-87 cells. Combined with our available microarray datasets from ALDOC knockdown models. We expected find the interaction partners, signaling pathways or metabolites by ALDOC regulated in only wild-type or R132 mutant of IDH1 in GBM. (please see line 193-200 of page 5 in revised manuscript.)

 How to explain the gender difference?

Answer: We calculated various clinicopathlogical factors, including age, gender, tumor grade and several molecular expression levels in our cohort. According our results, gender had no significant value in univariate analysis (p=0.105). Although gender revealed a boarding line in the multivariate analysis (p=0.045). However, the proportion of male/female in our cohort is unbalanced (n=40/24, respectively). Therefore, we will recruit more female glioma cases in our study to confirm if gender is an independent prognostic factor.

It is unclear why ALDOC is enriched in proneural subtype compared to mesenchymal subtype. Does ALDOC can inhibit GBM mesenchymal differentiation?

Answer: Thank you for reviewer’s suggestions. Recently, nuclear factor-κB (NF-κB) had been reported to play a possible role in mediating many of the central features associated with mesenchymal differentiation (Cells. 2018 Sep; 7(9): 125.). Moreover, NF-κB regulated several mesenchymal factors and further undergoes epithelial-mesenchymal transition (EMT). In this manuscript, we detected that ALDOC had higher express in the proneural than mesenchymal type. Therefore, we established the transcriptomics dataset with or without ALDOC knockdown in benign cells. Normalized by the Genespring software, we selected all probes that >1.5 fold change in knockdown ALDOC groups compared with vector control (n=689) to predict potential canonical pathways and upstream regulators through this signature. Our preliminary data showed that the knockdown of ALDOC up-regulated the subunits of NF-κB including RELA (p65), REL (c-Rel) and NFKB1 (p50). In addition, our data also revealed that TWIST1 was up-regulated in the ALDOC knockdown signature. In conclusion, we hypothesized that loss of ALDOC function may upregulate NF-κB activity and further induce EMT and mesenchymal differentiation in GBM. However, the detailed mechanism between ALDOC and NF-κB remained undetermined.

(Upper) The potential upstream regulators were been predicted by Ingenuity Pathway Analysis (IPA) software based on microarray from U87 shALDOC cells with a 1.5-fold change cut-off compare with vector control. (Lower) The network was be predicted based on the common signature that in the IPA database overlaid with microarray data from U87 shALDOC cells with a 1.5-fold change cut-off compare with vector control. The intensity of the node color indicates the degree of activate-(orange) and inhibit (blue) regulation following ALDOC knock down interactomics.

Round 2

Reviewer 1 Report

I would like to thank the reviewer for their quick and efficient reply. Regards

Author Response

I would like to thank the reviewer for their quick and efficient reply. Regards 

Answer: We are appreciated for your constructive recommendation.

Reviewer 2 Report

The authors have addressed the comments.

Author Response

The authors have addressed the comments.

Answer: We are appreciated for your constructive recommendation.

Reviewer 3 Report

The authors made some efforts to address reviewers' comments, but not all, especially for the in vivo validation. The authors just responded to reviewers, but did not add the revision data to the manuscript. I would suggest the authors to include all the revision data to the manuscript before the manuscript can be accepted.

Author Response

The authors made some efforts to address reviewers' comments, but not all, especially for the in vivo validation. The authors just responded to reviewers, but did not add the revision data to the manuscript. I would suggest the authors to include all the revision data to the manuscript before the manuscript can be accepted.

Answer: We are appreciated the kind reminder from reviewer #3. We added several results to the revised manuscript by these cell models and transcriptomics datasets. We also expanded our description in result 2.7. (please see the figure 7 and supplement figure 5-6, line 167-169, 183-198 in revised manuscript)
